# Dynamic Nomogram for Predicting Long-Term Survival in Terms of Preoperative and Postoperative Radiotherapy Benefits for Advanced Gastric Cancer

**DOI:** 10.3390/ijerph20032747

**Published:** 2023-02-03

**Authors:** Xinghui Li, Yang Yu, Cheng Zheng, Yue Zhang, Chuandao Shi, Lei Zhang, Hui Qiao

**Affiliations:** 1Cancer Institute of the General Hospital, School of Public Health and Management, Ningxia Medical University, Yinchuan 750004, China; 2Department of Epidemiology and Biostatistics, College of Public Health, Shaanxi University of Chinese Medicine, Xi’an 712046, China; 3Department of Neurosurgery, Huazhong University of Science and Technology Union Shenzhen Hospital, The 6th Affiliated Hospital of Shenzhen University, Shenzhen 518052, China; 4Guangdong Key Laboratory for Biomedical Measurements and Ultrasound Imaging, School of Biomedical Engineering, Shenzhen University Health Science Center, Shenzhen 518060, China; 5China-Australia Joint Research Center for Infectious Diseases, School of Public Health, Xi’an Jiaotong University Health Science Center, Xi’an 710061, China; 6Melbourne Sexual Health Centre, Alfred Health, Melbourne, VIC 3053, Australia; 7Central Clinical School, Faculty of Medicine, Nursing and Health Sciences, Monash University, Melbourne, VIC 3800, Australia; 8Department of Epidemiology and Biostatistics, College of Public Health, Zhengzhou University, Zhengzhou 450001, China

**Keywords:** advance gastric cancer, preoperative radiotherapy, postoperative radiotherapy, dynamic nomogram, LASSO, prognosis

## Abstract

Studies on the prognostic significance of preoperative radiotherapy (PERT) and postoperative radiotherapy (PORT) in patients with advanced gastric cancer (GC) remain elusive. The aim of the study was to evaluate the survival advantage of preoperative and postoperative radiotherapy and construct a dynamic nomogram model to provide customized prediction of the probability of prognostic events for advanced GC patients. We collected clinical records from 2010 to 2015 from the Surveillance, Epidemiology, and End Results (SEER) database with a specific target for stage II-IV GC patients treated with PERT or PORT. We used the least absolute shrinkage and selection operator (LASSO) regression model to identify factors that contribute to the overall survival (OS) of GC patients. The dynamic nomogram infographic was constructed based on the prognostic factors of tumor-specific survival. Out of the 3215 total patients (2271 [70.6%] male; median age, 61 [SD = 12] years), 1204 were in the PERT group and 2011 in the PORT group. Receiving PORT was associated with a survival advantage over PERT for stage II GC patients (HR = 0.791, 95% CI= 0.712–0.879, *p* < 0.001). The 1-, 3-, and 5-year OS rates were 89.9%, 63.8%, and 53.8% in the PORT group, whereas the corresponding rates were significantly lower in the PERT group (86.4%, 57.1%, and 44.3%, respectively, all *p* < 0.05). The survival prediction model demonstrated that patients aged > 65 years, with an advanced cancer development stage and tumor size >3 were independent risk factors for poor prognosis (all HR > 1, *p* < 0.05). In this study, a dynamic nomogram was established based on the LASSO model to provide a statistical basis for the clinical characteristics and predictive factors of advanced GC in a large population. PORT demonstrated significantly better treatment advantages than PERT for stage II GC patients.

## 1. Introduction

Gastric cancer (GC) is the fourth most common malignancy and cause of cancer-related death worldwide, with over one million new cases and an estimated 768,793 deaths in 2020 [1]. There is a high prevalence of advanced-stage presentation, comprising up to 70% of newly diagnosed GC patients at initial diagnosis in western countries [2]. Despite recent advances in the management of patients with GC over the past 20 years, the prognosis is still less than 12 months at advanced stages in the United States [3,4].

Gastrectomy is the primary treatment for resectable GC. However, even after complete resection of the lesion, 40% of patients still incur local and distant recurrences leading to death [5]. The prognosis is limited with surgery alone in GC patients with stages II and III, and the 5-year survival rate is only 20–50% [6,7,8]. Therefore, a reasonable choice is to add neoadjuvant and adjuvant therapy based on surgical resection, to improve the curative resection (R0) rate and reduce the local or regional recurrence rate. The INT0116 trial revealed significant survival benefits from adjuvant chemoradiotherapy (CRT) for GC patients after surgery, but this study was controversial because a large proportion of patients underwent D1 lymph node dissection, which is inferior to standard D2 lymph node dissection in terms of survival. [9]. Subsequently, the MAGIC trial in the UK explored the effects of the addition of perioperative chemotherapy on resectable esophageal and gastric adenocarcinoma patients [10] and showed that patients treated with perioperative chemotherapy had reduced tumor sizes and improved overall survival (OS) compared to those with surgery alone (36% vs. 23%, 5-year OS). This study promoted the formal inclusion of preoperative/perioperative chemotherapy in the NCCN guidelines. In Asian countries, the Korean ARTIST trial reported, in the subgroup analysis of lymph node-positive patients, 3-year disease-free survival rates in the chemotherapy group and concurrent CRT group of 72.3% and 77.5%, respectively (*p* = 0.036) [11]. However, the survival advantage of adjuvant or neoadjuvant radiotherapy as a treatment strategy has become increasingly controversial. To date, most studies have focused on the significance of adjuvant therapy in treatment, and comparisons of survival benefits between preoperative radiotherapy (PERT) and postoperative radiotherapy (PORT) are relatively limited.

Nomograms are useful for clinical services to predict the survival and prognosis of patients with a variety of tumors [12,13]. Conventional nomogram individualized prognosis prediction is based on regression models, such as logistic regression and Cox proportional risk regression, but they can only accommodate a relatively small number of covariates, and their prediction accuracies are questionable [14,15,16]. The use of least absolute shrinkage and selection operator (LASSO) regression models is a growing trend in GC prognostic studies, which can incorporate larger covariables in a large dataset, handle complex relationships between predictors and outcomes, and achieve high accuracy [17,18]. The few data explored are associated with PERT and PORT based on the survival of advanced GC patients. This study aimed to establish a dynamic nomogram model to provide customized prediction of the probability of prognostic events for advanced GC patients.

## 2. Materials and Methods

### 2.1. Study Population

The data of this study were retrieved from the Surveillance, Epidemiology, and End Results (SEER) program of the National Cancer Institute (NCI), which was a collaboration of 18 population-based regional cancer registries and the world’s largest publicly available cancer database. GC patients were selected directly using the SEER*Stat version 8.3.6 software, the SEER data did not contain any numerical symbols and are publicly available. Therefore, our study was not subject to ethical approval. We enrolled eligible patients according to the following criteria: (a) between 2010 and 2015, all patients were diagnosed with GC stage II-IV; (b) the primary site was the stomach; (c) underwent radical R0 gastrectomy and D1/D2 lymphadenectomy; (d) the exact treatment strategy could be traced (preoperative radiotherapy and postoperative radiotherapy); (e) pathologically diagnosed as GC. Ineligible cases with unknown or missing characteristic data were excluded. The patient selection flowchart of the current study is shown in Appendix A.

### 2.2. Study Variables

The variables were extracted from the SEER cohort (https://seer.cancer.gov/data/ (accessed on 20 March 2021)), and this dataset included patient demographics (sex, age at diagnosis, race, and marital status), pathologic characteristics (primary site, histologic type, tumor node metastasis (TNM) stages, tumor size, differentiation, summary stage, Lauren type, and bone, brain, liver, and lung metastases), comprehensive treatment (chemotherapy, RT, systemic therapy (e.g., targeted treatment), and follow-up data (follow-up duration and survival). Patients initially diagnosed with gastric cancer between 2010 and 2015 were selected for the study because information on the site of metastasis and comprehensive treatment were included in the database after 2010. As for the age, all cases were classified into ≤65 years and >65 years age groups; tumor size was regrouped into ≤3 cm, 3–5 cm, and >5 cm. The clinical TNM stage was based on the 7th American Joint Committee on Cancer stage system (AJCC). The primary site was divided into five different parts: gastric body, antrum/pylorus, lesser and greater curvature, antrum and pylorus, and others. The summary stage of the tumor was categorized into regional, distant, and localized. The histologic subtype was categorized as poorly differentiated, moderately differentiated, well undifferentiated, and undifferentiated. The pathological types were divided into adenocarcinoma and signet ring cell. Chemotherapy was grouped as “yes” or “no/unknown”, and radiation was grouped as PERT or PORT according to the SEER database. OS was the primary outcome, defined as data calculated from the date of diagnosis up to any cause of death or subsequent termination.

### 2.3. Statistical Analysis

Statistical analysis was performed using SPSS software version 23.0 (SPSS, Inc., Microsoft, Chicago IL, USA) and R software (version 3.6.1; https://www.R-project.org (accessed on 25 May 2021)). We used LASSO regression to analyze the data and screen out the optimal predictors among the present risk factors; GC patients were randomly divided into PERT group and PORT groups, and baseline characteristics were analyzed using the χ2 test or Fisher’s exact probability method. Multivariable regression analysis was performed to contract a predicting model by introducing the feature selected in the LASSO regression model. The survival curves that generate values of different variables were estimated via the Kaplan–Meier method, and the survival difference between groups was tested by performing a log-rank test. Statistically significant variables with *p* < 0.05 were entered into multivariate analysis based on the Cox regression model. Hazard ratios (HRs) and *p*-values with 95% confidence intervals (CIs) were used to estimate relative risk. Finally, all variables with statistical significance in multivariate analysis were incorporated into the nomogram visualization.

### 2.4. Dynamic Nomogram Construction

The nomogram prediction model performance evaluation included discrimination and calibration curves, which were assessed using a bootstrap method with 1000 resamples based on the original study cohort. Discrimination was evaluated using the concordance index (C index), which measured the accuracy with which the survival prediction model correctly predicted which patient would experience an event first for a randomly selected pair of patients. Calibration is another validation measure that evaluates and compares predicted survival with actual survival through calibration curves. The area under the receiver operating characteristic (ROC) curve provides good discrimination of the quality of the risk nomogram to distinguish true positives from false positives.

## 3. Results

### 3.1. Demographics and Clinical Characteristics of the Cohort

Between 2010 and 2015, 3215 total advanced GC patients were enrolled in the analysis after the exclusion criteria were applied (Appendix A); 2271 (70.6%) were males and 944 (29.4%) were females, and the median age was 61 (49–73) years. The number of patients with advanced GC who received PERT was 1204 (37.4%), and 2011 (62.6%) received PORT treatment. Table 1 summarizes the baseline clinicopathological characteristics of the patients in each treatment group in the original data and indicates the relevant differences between the two groups. Compared with patients in the PERT group, patients who received PORT were more likely to be female (742 [36.9%] vs. 202 [16.8%]; *p* < 0.001), and single or widowed (449 [22.3%] vs. 215 [17.9%]; *p* < 0.001). Patients in the PORT group had a greater frequency of poorly differentiated tumors (1403 [69.8%] vs. 638 [53.0%]; *p* < 0.001), a clinical stage of T4 (722 [35.9%] vs. 95 [7.9%]; *p* < 0.001), stage N2/3 (1137 [56.6%] vs. 350 [29.1%]; *p* < 0.001) and M1 (229 [11.4%] vs. 71 [5.9%]; *p* < 0.001), and a tumor size >5 cm (825 [41.0%] vs. 288 [23.9%]; *p* < 0.001) than those in the PERT group.

### 3.2. Feature Selection

We applied the LASSO regression algorithm to select the features in the treatment cohort, running K cross-validation 10 times for centralization and normalization of the inclusion variable based on the -2log likelihood and binomial family, and then selected the best lambda value. Sixteen potential predictors with non-zero coefficients were selected from all 18 relevant characteristic variables (Figure 1A), and each colored line represents a variable in the LASSO regression model. The most optimal tuning parameter lambda was 0.0018 when the partial-2log-likelihood binomial deviance reached its minimum value (Figure 1B).

### 3.3. Survival Outcomes

The 1-, 3-, and 5-year overall survival rates of stage-II GC patients were 89.9%, 63.8%, and 53.8% in the PORT group, which were significantly higher than 86.4%, 57.1%, and 44.3% in the PERT group, respectively (Appendix A, all *p* < 0.05). The survival outcomes of PERT versus PORT groups were evaluated. Compared with those in the PERT group, stage II, marital status (widowed), and tumor size (<3 cm) were shown to be significant predictors of good survival in the PORT group (Figure 2A,D,E, all *p* < 0.05). However, PORT did not show survival benefits over PERT for stages III and IV, distant-stage disease, and a primary site of the cardiac/fundus and body in advanced GC patients (Figure 2B,C,F–H, *p* > 0.05). The results of the Cox regression model are listed in Figure 3. In the multivariate Cox regression model, statistically significant covariates were age, stage T3-4, stage N2-3, M1 stage, and tumor size, which were independent risk factors for poor prognosis (HR > 1, *p* < 0.05). By contrast, chemotherapy and RT were found to be independent risk factors for poor prognosis (HR > 1, *p* < 0.05).

### 3.4. Nomogram Model Developed to Predict Survival

The dynamic nomograms of the 100th patient were constructed based on Regplot, including all eight independent factors from the multivariate Cox proportional hazards analysis (Figure 4). Constructed into eight different subgroups or different characteristic values of risk factors, all points were added up to calculate the probability of survival of different GC patients, to achieve an interactive function. The dynamic nomogram indicated that patients ≤ 65-years-old, with an early TNM stage, tumor size ≤ 3, and treated with chemotherapy and PORT were correlated with a better survival rate.

## 4. Discussion

The optimal therapy for advanced GC is perioperative multidisciplinary treatment, including chemotherapy, RT, immunotherapy, and targeted therapy [16]; therefore, comprehensive treatment is paramount to treatment selection. Biological differences between cancers from Western and East Asia countries add to the complexity of identifying standard treatment regimens based on several international trials [19,20]. In America and Europe, adjuvant chemotherapy combined with RT has been recommended as a standard strategy, as appropriate D2 lymphadenectomy is not commonly performed [21]. In contrast, postoperative chemotherapy based on randomized trials is more common in Asia [22]. The superiority of one treatment strategy over another cannot be ascertained.

Using a large GC dataset with a cross-sectional design, we demonstrated a significant association between the use of PORT and prolonged survival in patients diagnosed with pathologic stage II disease. Further, 1-, 3-, and 5- year overall survival rates were 89.9%, 63.8%, and 53.8% in the PORT group, which were consistently higher than those of 86.4%, 57.1%, and 44.3% in the PERT group. However, no significant difference in overall survival was found between patients with stage III and stage IV disease who completed PERT and PORT for advanced GC; subsets of individuals might not benefit more from PORT owing to poor tolerance and limited toxicity.

Multiple trials have shown that preoperative adjuvant therapy for GC patients enhances the rate of a complete pathologic response, downstages advanced tumors, eliminates possible micrometastases, and is associated with margin-negative resection along with overall survival [23,24,25]. However, the survival time of patients with stage II GC has not been improved by PERT. The reasons for PERT being ineffective for GC patients are manifold. First, a possible reason was that with PORT, appropriate patients can be selected according to the postoperative pathological stage, to avoid unnecessary radiotherapy for patients without indications for radiotherapy [26,27]. Second, the tumor burden of PORT was lower than that of PERT; PORT is mainly targeted at subclinical lesions, and the efficacy of PORT is better for GC patients with stage II disease [28].

Limited data regarding postoperative therapies are available. Our findings were echoed in other research. The US INT0116 trial and the Dutch CRITICS trial [29] both suggested that PORT is effective for patients with specific treatment modalities and disease stages. Meanwhile, only 64% of patients in INT0116 completed postoperative treatment, and 17% of patients were unable to complete radiotherapy. Similarly, 50% of patients in the CRITICS cohort were able to complete postoperative chemotherapy or chemoradiotherapy. However, in our study, only patients who completed PERT or PORT treatment were included in the cohort, which makes our conclusion more credible. A recent study—the Korean ARTIST-2 trials—showed that PORT following D2 gastrectomy is associated with no additional benefit compared to that with chemotherapy alone [30], but for patients with pathologically positive lymph nodes, PORT demonstrated a survival advantage with no statistical significance (*p* = 0.38). It may be due to regional differences, the degree of lymph node dissection, tumor stage, and the wide use of postoperative chemotherapy that the results of these studies are inconsistent.

We demonstrated that advanced GC patients had significantly greater odds of having poor prognosis when they showed the five factors as follows: age > 65 years, T3-4 stage, N1-3 stage, M1 stage, and tumor size > 3 cm. However, PORT was still significantly associated with favorable cancer-specific survival (HR < 1, *p* < 0.001). We have employed the LASSO regression for feature selection and the multivariate Cox regression model to reduce confounding bias caused by this study being retrospective in nature, and our findings remained valid with these multivariate models. Elderly cases are frequently associated with restricted inclusion in clinical trials owing to the physiological changes that occur with age, including pharmacodynamic variability, diminished organ functions, and impaired functional status, which necessitate individualized treatment approaches. One study reported survival advantages for patients less than 70 years old receiving adjuvant CRT, but not in those above 70-years-old [31]. Tumor size is also one of the important prognostic factors in patients with GC, which is significantly related to tumor progression, lymph node metastases, and recurrence. Patients with large tumors tend to have more aggressive features and a worse prognosis than patients with small tumors [32]. 

In this study, the dynamic nomogram OS prediction model took into account eight indicators after the screening based on multivariate analysis, and an AUC = 0.798 indicated favorable discrimination and calibration ability in the cohort. Estimating survival probability based on the TNM stage alone is not always accurate. The dynamic nomogram model appropriately illustrates how prognosis changes significantly with other factors, such as patient age, TNM stage, tumor size, PERT, and PORT. With the collection of more specific patient and tumor data, such as genetic information and molecular tumor biomarkers, the use of these types of predictive models will become increasingly important.

We successfully constructed a dynamic nomogram to control for the bias associated with patients with GC after surgical resection. Nevertheless, some limitations are worth mentioning. First, this study was conducted retrospectively, making it vulnerable to the biases of this type of study format. The result needs to be further confirmed based on prospective studies with western datasets. Second, detailed prognostic information was not included in the SEER database, such as the proportion of D2 lymph nodes removed, RT field of vision, RT dose information, chemotherapy protocol, specific treatment, and patient’s personal history, and thus, selection bias might exist. Third, we only employed dynamic nomogram approaches for categorical outcome prediction, and additional approaches may be explored as part of a future investigation. Fourth, our study represents concurrent predictions of outcomes and features from the same cross-section and is not a prospective prediction, which will be part of future studies.

## 5. Conclusions

In conclusion, this large cohort from the SEER database revealed that PORT is associated with survival benefits for GC patients with advanced-stage compared to PERT. It is worth noting that PORT was more likely to be beneficial for patients with stage II disease. Furthermore, the variables age, race, and chemotherapy were found to be confounding factors affecting prognosis in the advanced stage based on the dynamic nomogram model. The dynamic nomogram model is a useful tool for contending with confounding and selection issues that can be used to properly conduct high-quality research.

## Figures and Tables

**Figure 1 ijerph-20-02747-f001:**
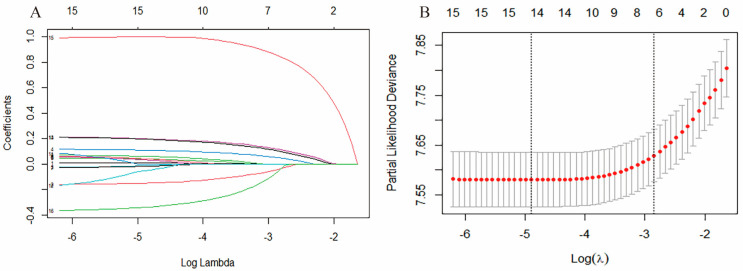
(**A**) LASSO coefficient profiles of the candidate predictors. (**B**) Selection of the optimal penalization coefficient in the LASSO regression.

**Figure 2 ijerph-20-02747-f002:**
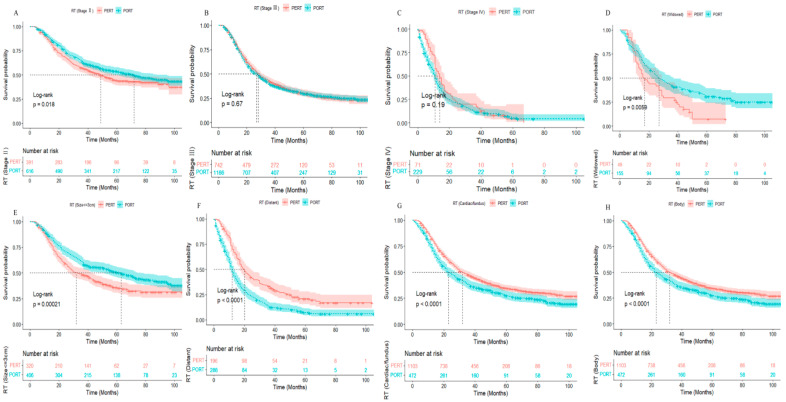
Kaplan–Meier curves for overall survival based on treatments and clinically related factors in advanced GC patients. (**A**–**C**) Survival curves of RT in different stages, stage II (**A**), stage III (**B**), and stage IV (**C**), marital status (widowed) (**D**), tumor size (<3 cm) (**E**), summary stage (distant) (**F**), primary site (Cardiac/fundus) (**G**) and (body) (**H**) between PERT and PORT group in advanced GC patients. GC: gastric caner; RT: radiotherapy; PERT: preoperative radiotherapy; PORT: postoperative radiotherapy.

**Figure 3 ijerph-20-02747-f003:**
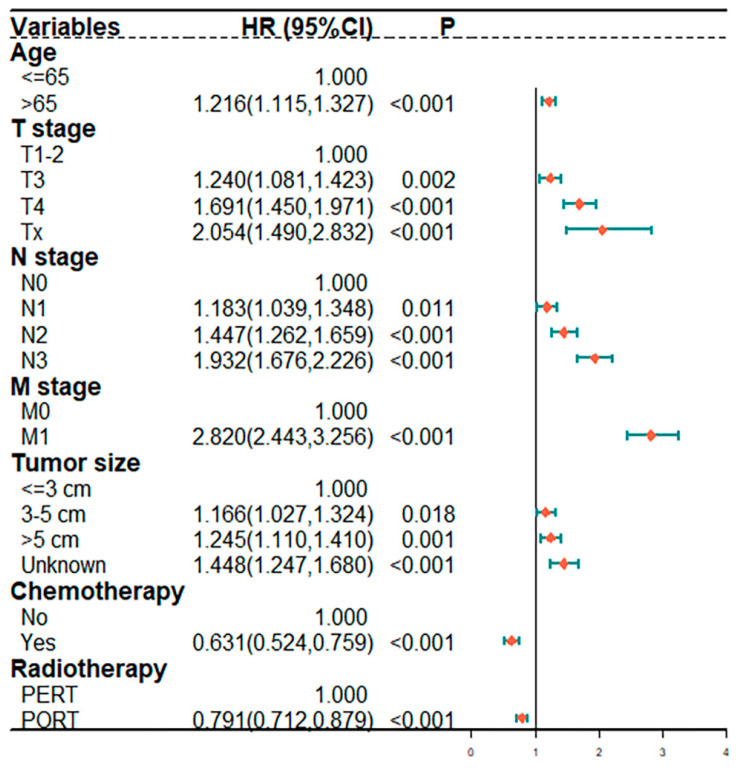
Multivariate analysis for factors associated with OS.

**Figure 4 ijerph-20-02747-f004:**
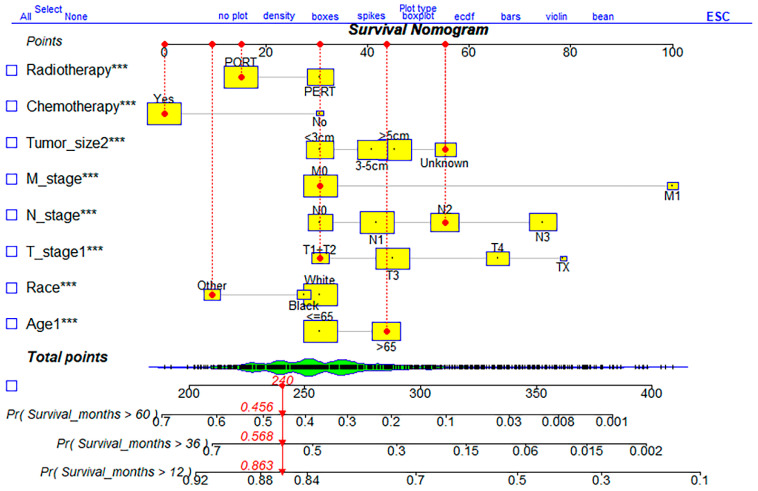
The dynamic nomogram for predicting 1-, 3- and 5-year OS of advanced GC patients. OS: overall survival; GC: gastric cancer.

**Table 1 ijerph-20-02747-t001:** Baseline demographic and clinical characteristics of patients with gastric cancer.

Characteristic	Total (*n* = 3215)	PERT (*n* = 1204)	PORT (*n* = 2011)	*p*
Age				0.309
≤65	1902 (59.2)	726 (60.3)	1176 (58.5)	
>65	1313 (40.8)	478 (39.7)	835 (41.5)	
Sex				<0.001
Male	2271 (70.6)	1002 (83.2)	1269 (63.1)	
Female	944 (29.4)	202 (16.8)	742 (36.9)	
Race				<0.001
White	2278 (70.9)	1062 (88.2)	1216 (60.5)	
Black	377 (11.7)	50 (4.2)	327 (16.3)	
Other	560 (17.4)	92 (7.6)	468 (23.3)	
Marital status				<0.001
Married	2097 (65.2)	822 (68.3)	1275 (63.4)	
Divorced/Separated	313 (9.7)	128 (10.6)	185 (9.2)	
Single	460 (14.3)	166 (13.8)	294 (14.6)	
Widowed	204 (6.3)	49 (4.1)	155 (7.7)	
Unknown	141 (4.4)	39 (3.2)	102 (5.1)	
Primary site				<0.001
Cardiac/fundus	1575 (49.0)	1103 (91.6)	472 (23.5)	
Body	219 (6.8)	15 (1.2)	204 (10.1)	
Antrum/pylorus	703 (21.9)	16 (1.3)	687 (34.2)	
Lesser/greater curvature	356 (11.1)	30 (2.5)	326 (16.2)	
Other	362 (11.3)	40 (3.3)	322 (16.0)	
Histology				<0.001
Adenocarcinoma	2896 (90.1)	1151 (95.6)	1745 (86.8)	
Other	319 (9.9)	53 (4.4)	266 (13.2)	
TNM Stage				<0.001
II	1007 (31.3)	391 (32.5)	616 (30.6)	
III	1908 (59.3)	742 (61.6)	1166 (58.0)	
IV	300 (9.3)	71 (5.9)	229 (11.4)	
T stage				<0.001
T1-2	478 (14.9)	178 (14.8)	300 (14.9)	
T3	1857 (57.8)	926 (76.9)	931 (46.3)	
T4	817 (25.4)	95 (7.9)	722 (35.9)	
Tx	63 (2.0)	5 (0.4)	58 (2.9)	
N stage				<0.001
N0	623 (19.4)	284 (23.6)	339 (16.9)	
N1	1105 (34.4)	570 (47.3)	535 (26.6)	
N2	773 (24.0)	265 (22.0)	508 (25.3)	
N3	714 (22.2)	85 (7.1)	629 (31.3)	
M stage				<0.001
M0	2915 (90.7)	1133 (94.1)	1782 (88.6)	
M1	300 (9.3)	71 (5.9)	229 (11.4)	
Differentiation				<0.0001
Poorly	2041 (63.5)	638 (53.0)	1403 (69.8)	
Moderately	783 (24.4)	382 (31.7)	401 (19.9)	
Well	82 (2.6)	43 (3.6)	39 (1.9)	
Undifferentiated	309 (9.6)	141 (11.7)	168 (8.4)	
Summary stage				0.010
Localized	241 (7.5)	108 (9.0)	133 (6.6)	
Regional	2490 (77.4)	900 (74.8)	1590 (79.1)	
Distant	484 (15.1)	196 (16.3)	288 (14.3)	
Lauren type				<0.001
Intestinal	333 (10.4)	60 (5.0)	273 (13.6)	
Diffuse	252 (7.8)	28 (2.3)	224 (11.1)	
Mixed	100 (3.1)	21 (1.7)	79 (3.9)	
Unknown	2530 (78.7)	1095 (90.9)	1435 (71.4)	
Tumor size				<0.001
≤3 cm	726 (22.6)	320 (26.6)	406 (20.2)	
3–5 cm	919 (28.6)	347 (28.8)	572 (28.4)	
>5 cm	1113 (34.6)	288 (23.9)	825 (41.0)	
Unknown	457 (14.2)	249 (20.7)	208 (10.3)	
Bone metastases				<0.153
Yes	129 (4.0)	56 (4.7)	73 (3.6)	
No/Unknown	3086 (96.0)	1148 (95.3)	1938 (96.4)	
Brain metastases				<0.001
Yes	39 (1.2)	4 (0.3)	35 (1.7)	
No/Unknown	3176 (98.8)	1200 (99.7)	1976 (98.3)	
liver metastases				0.142
Yes	72 (2.2)	21 (1.7)	51 (2.5)	
No/Unknown	3143 (97.8)	1183 (98.3)	1960 (97.5)	
lung metastases				0.820
Yes	31 (1.0)	11 (0.9)	20 (1.0)	
No/Unknown	3184 (99.0)	1193 (99.1)	1991 (99.0)	
Chemotherapy				<0.001
Yes	3061 (95.2)	1193 (99.1)	1868 (92.9)	
No/Unknown	154 (4.8)	11 (0.9)	143 (7.1)	

## Data Availability

The analysis in this paper is based on the SEER database. The article/Appendix A contains the original contributions presented in the research report. Further enquiries may be directed to the corresponding author.

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
