# Peer review of "Dynamic Nomogram for Predicting Long-Term Survival in Terms of Preoperative and Postoperative Radiotherapy Benefits for Advanced Gastric Cancer"

_ijerph, 2023, doi:10.3390/ijerph20032747_

Round 1

Reviewer 1 Report

The authors deal with an important issue of gastrointestinal oncology, namely whether giving preoperative or postoperative radiotherapy (PERT vs. PORT) to advanced gastric cancer (GC) patients. The topic is interesting and important, the statistical work is spectacular, however, the reviewer feel that over the epidemiology/statistical results, the real oncological message is under-emphasized. Similar works could clarify some clinical doubts, e.g., the role of RT in the care of locoregional advanced GC. Since in the latest years combined chemotherapy regimens became the standard of care in the preoperative (and postoperative) setting, it is important to prove that RT may have any role in the postoperative setting in discrete situations (e.g., incomplete resection, R1-R2 or in Stage II (III?) where D2 dissection was not reached.). On the other hand, in Stage III (IV) cases -supposing high chance for dissemination - the basic treatment category may be chemotherapy (or other systemic treatments). Conclusively, I missed some general (and theoretical) messages/conclusions for the clinician readers.     

Beyond these general thoughts, there are a great number of inattentions in the text as well.

Summarizing these findings, I suggest major revision and resubmission, considering the importance and the great work in the background. My concerns/ findings/ recommendations are the following.

1, Title: The authors use „perioperative” and „postoperative” phrases, nonetheless, the application of perioperative RT is a special and very rare indication and not a routine one.

2, Abstract: - Lines 28-29: „This study aimed quantified the survival benefit of adjuvant radiotherapy for advanced GC patients” ….over PERT…over nonRT strategies? Please clarify.

- Line 30: The authors collected clinical records from 2010 to 2015 (see Material and Methods section as well). However, in the Results section and in the Supplementary Material the examined period was signed between 2000 and 2018…. Please clarify it.

- Line 31 (and later on in the Materials and Methods section as well): „stage II-IV GC”…….Maybe Stage IVa……Metastatic GC is not an indication for PERT nor PORT, moreover oligometastatic GC is not a frequent clinical finding. There are only limited number of Stage IV (metastatic) cases, but please explain their role in the material. (The authors did not describe the delivered RT doses, maybe these cases were palliative RT courses?)

- Line 35: median/average age?

- Lines 36-37: „PORT was significantly associated with more advanced GC progression in stage II compared with PERT” Please clarify this statement.

3, Introduction: - Lines 53-55: „Despite recent advances in the management of patients with GC over the past 20 years, the prognosis is still less than 12 months in the United States.”  It is not true, maybe in metastatic/ relapsed diseases.

- Lines 57-58:  „there were still 40% of patients incurred local and distant recurrences leading to death”…. Please clarify this statement.  

(I recommend a distinct paragraph in the Introduction or in the Discussion section to talk about the role of chemotherapy, since currently CT is the base of perioperative treatment of GC. I also recommend a paragraph to note the present indication lists for PERT and PORT. Nevertheless, there are differences in the indications of PERT and PORT, and there could be some special clinical circumstances, so the comparison is not an easy task.)

- There is a confusion in the sequence of references, I found a great number of wrong numbers in the brackets (see line 65, 73 etc.). Please overview all the references once more.

4, Materials and Methods: - Line 99: „the stomach was the primary site of infection”….Please clarify it.  

- Line 105: I missed the categories of R0-R1-R2, nevertheless this finding is important to determine the necessity of PORT.  Please correct or explain (missing data) it.

- Line 109: CT, RT and any other systemic therapy (e.g. targeted treatment)…

- Line 121: What are PECT and POCT? Please clarify them.  

5, Results: - Lines 158-162: The pathological features of the tumors in the PORT group were riskier, however the survival results of PORT were significantly better, than the results of PERT. Please explain it the Discussion section….. (These differences also ground the difficulty to compare the results of PERT and PORT.)  

- Line 177: To be a widow is a favorable prognostic factor? If it possible, please elucidate it.

- Lines 178-179: “However, PORT demonstrated no significant advantage over PECT in stage III, IV and distant GC patients” … It is really a very important message of the work. I suggest emphasizing it.  

- I suggest a more detailed descriptions in Figure Legends, mainly by Figures 3. and 4. Moreover, Figure 4. is not clearly visible.

- Table 1 (and Table S1): What is the significance of Summary Stages (regional, distant, localized)? The detailed work uses the conventional Stage categories. ….Please explain it in the text.

5, Discussion: - Lines 227-228: The possible reason for a great number of interrupted RT courses in the classical RT studies could have been the utilization of old RT techniques (and high toxicity, ineffective supportive care etc.). Currently we use more sophisticated technology. Please clarify it…..Why was not the treatment intention the base of the research? I accept it, however, please explain it.      

6, Supplementary Materials: - Figure 1: The exclusion criteria are visible in 2 places. Please correct it. Note: there is a great number of blank data. Is it not a difficulty?

6, Conclusion: „PORT was more likely to benefit patients who were widowed, in advance stage of cancer development (T3/N2)”…Please explain it, if it possible.

Author Response

Dear eidtor,

Thank you very much for giving us the opportunity to revise the manuscript. In this revision, we have revised the text and prepared the files according to the reviewers' suggestions and provided a point-by-point response. We have made extensive corrections on the latest rrevision.

We provide both a clean version and a tracked-change version for your reference.

Best regards

Prof. Lei Zhang

China-Australia Joint Research Center for Infectious Diseases, School of Public Health, Xi'an Jiaotong University Health Science Center, Xi'an, Shaanxi, China.

Prof. Hui Qiao, M.D.,

School of Public Health and Management, Ningxia Medical University, Yinchuan, Ningxia, 750004 China.

Reviewer 2 Report

29: The aim of the study is to quantify the benefit of Adjuvant Radiotherapy in CG using a dynamic nomogram. Almost 2/3 of the sample population in the study underwent neoadjuvant radiotherapy. The sentence is misleading

62 – 65 : One of the criticisms of the INT116 study is that a large proportion of patients underwent D1 lymph node dissection, which is inferior to standard D2 lymph node dissection in terms of survival. It has been suggested that this sample population derived the greatest benefit from PORT and it should be clear to the reader that the effect size would be different if all patients underwent D2 LN dissection. The authors statement is misleading if they don’t comment on the quality of surgery.

Figure 2: The description doesn’t correspond to the related diagrams, needs correction

71: Correct “Trail” with trial (Artist trial)

99: “the stomach was the primary site of infection” should be corrected

99: The authors should explain why stage IV of GC was included in the study since RT is given in stage IV for palliative reasons and at lower doses. Moreover, it should be clarified if patients underwent radiation therapy for palliative or radical treatment and information of the dose should be known. If there is no information on the RT dose, it should be stated. The study aims to quantify the survival benefit of adjuvant RT in GC and metastatic GC does not fall into this category.

179: what is the difference between stage IV and “distant GC patients”

Table 1 In the histology category, signet ring cells are also adenocarcinomas. Adenocarcinoma vs signet ring categories should be reconsidered

Table 1, Summary stage: categories should be placed in order (localised, regional, distant)

240: Correct “staing”

276: authors should provide an explanation to why widowed patients were more likely to benefit from PORT

Author Response

(The authors gave the same response as above.)

Round 2

Reviewer 1 Report

I accept the answers and the corrections of the authors. However,I realized 3 remnant mistakes: 

1, Title: please use preoperative and not perioperative

2, In the Results section the collection period is 2000-2018. Please correct it.

3, In the supplementary Table 1: there is two rectangles about the blank data exception. Please correct it.  

Author Response

Dear eidtor,

Thank you very much for giving us another chance to revise the manuscript. In this revision, we modified the text and prepared the files according to the reviewers' suggestions and provided a point-by-point response. We have made amedment on the latest rrevision.

We provide both a clean version and a tracked-change version for your reference. Please see the attachment.

Best regards

Prof. Lei Zhang

China-Australia Joint Research Center for Infectious Diseases, School of Public Health, Xi'an Jiaotong University Health Science Center, Xi'an, Shaanxi, China.

Prof. Hui Qiao, M.D.,

School of Public Health and Management, Ningxia Medical University, Yinchuan, Ningxia, 750004 China.
